# Illuminating the Invisible: Fluorescent Probes as Emerging Tools for Micro/Nanoplastic Identification

**DOI:** 10.3390/ijms262311283

**Published:** 2025-11-21

**Authors:** Junhan Yang, Kaichao Zheng, Weiqing Chen, Xiaojun Zeng, Yao Chen, Fengping Lin, Daliang Li

**Affiliations:** Fujian Key Laboratory of Innate Immune Biology, Biomedical Research Center of South China, College of Life Sciences, Fujian Normal University, Fuzhou 350117, China; 108032024047@student.finu.edu.cn (J.Y.); qsz20231736@student.fjnu.edu.cn (K.Z.); qsz20241770@student.fjnu.edu.cn (W.C.); qsx20241328@student.fjnu.edu.cn (X.Z.); 108122022044@student.fjnu.edu.cn (Y.C.); fplin@fjnu.edu.cn (F.L.)

**Keywords:** microplastics, nanoplastics, fluorescence probes, environmental monitoring, solvatochromism, aggregation-induced emission, plasmon-enhanced fluorescence, specific recognition

## Abstract

The pervasive environmental contamination by micro- and nanoplastics (MNPs) presents a formidable analytical challenge, necessitating the development of rapid and sensitive detection methods. While conventional techniques often suffer from limitations in sensitivity and throughput, fluorescent probe-based technology has emerged as a powerful alternative. This review charts the evolution of these probes, from initial stains relying on hydrophobic adsorption to advanced molecular designs engineered for specific chemical recognition. We critically examine key operational mechanisms, including the solvatochromic response of Nile Red, polarity-discriminatory probes enabling a “microplastic rainbow,” and targeted systems achieving turn-on fluorescence via restriction of intramolecular rotation. Furthermore, we highlight cutting-edge signal enhancement strategies, such as plasmon- and metal-enhanced fluorescence, which amplify detection to the femtogram level. Special emphasis is placed on the distinct challenges posed by nanoplastics, including their propensity for aggregation in aqueous matrices that exacerbates false positives and their superior ability to breach biological barriers, and how AIE luminogens and PEF/MEF strategies mitigate these issues through enhanced signal-to-noise ratios and subcellular resolution, differing from their application to microplastics. Critically, we address the imperative for low-toxicity probe designs, emphasizing biocompatibility and biodegradability criteria to facilitate safe, long-term in vivo tracking and widespread ecological surveillance. The integration of these sophisticated probes with smart, “activate-on-target” systems is paving the way for next-generation MNP analysis, offering critical insights for environmental monitoring and toxicological assessment.

## 1. Introduction

Global plastic waste generation has doubled over the last two decades, with the majority entering ecosystems [1]: a crisis exacerbated by the COVID-19 pandemic’s surge in single-use plastics [2,3]. While visible debris is concerning, a more insidious threat arises from its progressive fragmentation into micro- and nanoplastics (MNPs) under environmental stressors [4,5]. This degradation cascade produces secondary MNPs, which, alongside deliberately manufactured primary MNPs [6], persist indefinitely in the environment, evading capture and permeating ecosystems with alarming efficiency [7,8,9]. Owing to their small size and heterogeneous nature, MNPs have become pervasive contaminants, infiltrating terrestrial, aquatic, and atmospheric systems globally [10,11,12]. Their transport, governed by hydrodynamic and aeolian forces, enables transcontinental dispersion, while their variable size, polymer composition, and morphology enhance environmental mobility and complicate remediation [13,14,15,16].

This ubiquity translates into direct ecological and health risks. Mounting evidence indicates that MNP exposure induces organismal dysregulation, including oxidative stress, gut microbiome dysbiosis, and impaired reproduction [17,18]. Notably, nanoplastics demonstrate enhanced bioinvasive potential, enabling subcellular translocation and damage [19,20]. Chronic exposure may even precipitate novel pathologies like Plasticosis, a disease characterized by persistent inflammation and tissue fibrosis [21,22,23,24]. These cascading impacts, arising from the pervasive global cycle of MNPs from sources to biological exposure (Figure 1), necessitate robust analytical tools to assess risk and guide mitigation [25]. Recent evidence highlights MPs/NPs’ potential to contribute to human carcinogenesis through mechanisms like DNA damage, oxidative stress, inflammation, and autophagy dysregulation, with bioaccumulation in organs such as the liver, lungs, and brain [26]. Fluorescent probes, by enabling real-time tracking of MNP uptake and cellular interactions, are crucial for elucidating these toxicological pathways and informing public health strategies.

Nanoplastics (NPs), defined as particles below 100 nm, present additional analytical hurdles that diverge markedly from those of microplastics (MPs). Their diminutive size promotes spontaneous aggregation in aqueous environments, driven by van der Waals forces and surface charge interactions, which can lead to overestimation of particle counts and false positives during detection [27]. Moreover, NPs exhibit heightened bioinvasive capabilities, readily traversing cellular membranes and blood–tissue barriers via endocytosis or passive diffusion, thereby amplifying toxicological concerns such as neurotoxicity and systemic inflammation [28]. Conventional probes often falter here, as aggregation artifacts obscure signals, whereas advanced modalities like AIE luminogens exploit restricted intramolecular motion to yield aggregation-resistant fluorescence enhancement, enabling precise NP tracking in vivo. Similarly, PEF and MEF harness plasmonic amplification to resolve NPs at subcellular scales, contrasting with their broader application to MPs, where size-based filtration suffices.

Accurately characterizing these pollutants is therefore paramount but fraught with complexity [29]. Diverse polymer types of MNPs and their residence in complex environmental matrices demand sophisticated, multi-stage sample preparation and analysis [30,31,32,33,34]. While a suite of technologies exists—from vibrational spectroscopy for identification to chromatography for composition—they often grapple with limitations in sensitivity, throughput, and cost for routine trace-level analysis [35,36,37,38]. Within this analytical landscape, fluorescence microscopy has emerged as a significant technique for rapid, high-resolution characterization [39,40]. Its power derives from fluorogenic probes, the photophysical properties of which modulate upon association with MNPs, enabling visualization and providing diagnostic signatures for polymer composition [41,42,43,44,45].

The operational principles of fluorescent probes for MNP characterization have undergone significant evolution, establishing the structure of this review. Initially, detection relied on nonspecific hydrophobic adsorption, utilizing the fluorescence turn-on of solvatochromic dyes. A paradigm shift toward chemical recognition enabled polymer differentiation through ratiometric responses and, most notably, target-specific fluorogenesis via mechanisms like restriction of intramolecular rotation. Currently, the field is integrating chemical specificity with physical amplification using plasmonic substrates and smart, “activate-on-target” materials. This trajectory—from general staining to specific recognition and integrated enhancement—guides the development of next-generation tools for MNP analysis.

## 2. Basic Detection Strategy: Fluorescence Staining Based on Hydrophobic Adsorption

Early methods for microplastic (MP) detection primarily leveraged the hydrophobic affinity of organic dyes for plastic surfaces, utilizing simple staining to enhance visibility [46,47]. These techniques are operationally simple and low-cost, establishing the fundamental approach for the rapid, initial screening of MPs in environmental samples. The most prominent probe in this class is Nile Red (NR), a solvatochromic dye that exhibits a strong “turn-on” fluorescent response when adsorbed onto non-polar polymer surfaces, such as polystyrene (PS), polyethylene (PE), and polypropylene (PP) [42,48]. This property makes NR a valuable, user-friendly screening tool for diverse aqueous samples. However, its major drawback is its inherent lack of specificity; the hydrophobic interactions driving its binding are not unique to synthetic plastics, often leading to significant false positives from naturally occurring lipophilic organic matter, including wood fragments or chitin [49]. Furthermore, dye leaching poses a risk, potentially causing misinterpretation in biological uptake studies where detached dye might be mistaken for internalized particles [50]. Alternatives within this category, such as Coumarin 6, offer improved cost-efficiency, being significantly less expensive than Nile Red [8]. Coumarin 6 has also been shown to provide more consistent and uniform staining for certain polymers like polyvinyl chloride (PVC) and polyamide (PA) [51,52]. Despite these advantages, it suffers from the same fundamental issue of non-specific binding. Another common hydrophobic dye, Rhodamine B (RhB), offers good water solubility and achieves rapid, stable staining. However, its binding is weak and highly reversible, with high reported desorption rates (79–89%), and its classification as a potential carcinogen raises serious biotoxicity concerns for environmental and biological tracing applications. The collective limitations of these first-generation hydrophobic dyes—namely their non-specificity, potential for leaching, and, in the case of RhB, toxicity—underscore the critical need for next-generation probes. These advanced systems must move beyond generic hydrophobicity to exploit specific chemical interactions for more accurate and reliable microplastic identification 

### 2.1. Nile Red: The Most Widely Used Hydrophobic Probe

Nile Red (NR) stands as the preeminent solvatochromic dye in microplastic (MP) analysis, a status earned through its operational simplicity, cost-effectiveness, and pronounced fluorescence “turn-on” response upon partitioning into hydrophobic polymer matrices [53,54] (as shown in Figure 2B). Its detection mechanism is governed by a profound polarity shift: the dye exhibits weak fluorescence in aqueous environments but undergoes a significant enhancement in quantum yield and a characteristic redshift in emission upon interacting with the non-polar surfaces of plastics like polystyrene (PS) and polyethylene (PE) [53,54,55].

Protocol refinement has been pivotal to its success. The formulation of NR in a non-polar solvent such as n-heptane (NR-H) optimizes dye partitioning, enabling direct, in situ staining in complex aqueous matrices—from wastewater to seawater—without demanding extensive pre-processing (as shown in Figure 2A). This capacity for rapid screening, coupled with validation from techniques like SEM and DLS confirming preserved particle morphology, solidifies NR’s role as an unparalleled tool for initial contamination assessment [53].

However, the very hydrophobicity that underpins NR’s utility is also the source of its principal limitation: a fundamental lack of specificity. The dye’s non-specific adsorption does not discriminate between synthetic polymers and ubiquitous natural organic matter (OM) like chitin or cellulose, leading to substantial false positives [56,57]. This challenge is compounded by other critical shortcomings, including: (i) inconsistent staining across polymer types (e.g., poor performance on rigid polyvinyl chloride (PVC)) and within individual particles, which complicates quantification [56]; and (ii) the potential for dye leaching, a critical confounder in biological uptake studies where free dye can be misinterpreted as particle internalization [56].

In response to these challenges, the scientific community has systematically refined NR-based methodologies into sophisticated and standardized workflows. This evolution is built upon three pivotal advancements: First, the implementation of system-specific calibration establishes a quantitative pixel brightness (PB) threshold for each microscope, objectively discriminating target signals from background and false positives [56]. Second, morphological verification via bright-field microscopy, guided by identification keys, conclusively confirms the synthetic nature of pre-screened candidates [56] (Figure 2C). Finally, for bio-imaging, advanced strategies like the encapsulation of fluorophores within the polymer matrix during synthesis prevent leaching, ensuring signal fidelity.

In conclusion, while Nile Red has irrevocably shaped the field by providing an accessible entry point for MP detection, its enduring legacy may lie in how its inherent limitations catalyzed a critical evolution in analytical philosophy—away from reliance on a single method and toward the adoption of robust, multi-modal workflows. This acknowledgment of NR’s constraints not only enhances current data reliability but also clearly delineates the need for the next generation of specificity-driven chemical probes, thereby setting the stage for the advanced molecular designs discussed subsequently.

### 2.2. Coumarin 6 (C_20_H_18_N_2_O_2_S): Economical Alternative to Nile Red

Building upon the foundation of solvatochromic probes like Nile Red, Coumarin 6 (Figure 3A) has recently emerged as a promising and cost-effective alternative for the fluorescence staining of microplastics (MPs) [58]. As a lipophilic dye, it operates on a similar principle: its fluorescence intensity is significantly enhanced upon partitioning from a polar aqueous environment into the hydrophobic domains of plastic polymers, a classic solvatochromic “turn-on” response with an emission peak around 500 nm.

The primary advantage of Coumarin 6 lies in its compelling balance of performance and practicality. Studies have demonstrated its effectiveness across a wide spectrum of common polymers, including polyethylene, polypropylene, polystyrene, and PVC (as visualized by their distinct spectral profiles in Figure 3B). While offering a more uniform emission profile and consistent staining compared to the sometimes-variable performance of Nile Red, its most distinctive feature is a dramatically lower cost, enhancing its accessibility for large-scale monitoring campaigns.

Methodologically, Coumarin 6 staining is straightforward. It typically involves immersing MP samples in a dye solution (e.g., ~1 mg L^−1^ in an acetone-ethanol mixture) for a short period, enabling rapid and direct staining without complex processing. This protocol has been successfully validated in environmental matrices such as seawater, where it exhibited a strong correlation between particle area and fluorescence intensity (R^2^ > 0.92) (Figure 3C), facilitating semi-quantitative analysis. Furthermore, a significant practical benefit is its compatibility with downstream confirmatory techniques like FTIR, allowing for a dual-modality workflow where Coumarin 6 serves as an efficient pre-screening tool.

Despite these strengths, Coumarin 6 inherits the fundamental limitation of hydrophobic adsorption-based probes: a lack of absolute specificity. It can yield false positives from natural organic particles with inherent hydrophobicity or autofluorescence, necessitating complementary verification, typically via morphological inspection. Nevertheless, by offering a reliable, low-cost, and rapid staining option, Coumarin 6 effectively bridges the gap between sophisticated instrumentation and simple visual assessment, establishing itself as a valuable asset within the standard MP analysis toolkit.

## 3. Towards Precise Identification: Advanced Probes with Selectivity

To overcome the non-specificity of first-generation stains, advanced fluorescent probes are increasingly engineered for precise polymer identification via targeted chemical interactions. This paradigm is broadly characterized by two complementary strategies: chemical differentiation, which deciphers polymer properties like polarity into identifiable signal patterns, and target-specific recognition, which employs molecular “lock-and-key” mechanisms for exclusive labeling. The solvatochromic probe DANS serves as a prime example of the first strategy, creating a “microplastic rainbow” for visual discrimination. Building on this foundational concept, the DPNA probe demonstrates the power of the second approach, achieving a turn-on, near-infrared fluorescence signal specific to polyurethane through a directed hydrogen-bonding mechanism. While these advances mark a significant leap forward, they also bring practical challenges to the fore, such as the significant biotoxicity of probes like PTSA, underscoring the critical need for parallel development of environmentally safe staining methodologies.

Developing polymer-specific “turn-on” probes for chemically inert polymers such as PE and PET remains challenging because these materials lack strongly polar or reactive surface groups. Nevertheless, several feasible chemical strategies exist, including molecularly imprinted polymer (MIP) recognition layers that generate selective binding cavities matching polymer oligomers or surface motifs. Additionally, probes that react with characteristic weathering products—such as carbonyl or hydroxyl groups produced during photooxidation—could selectively activate on aged PE/PET while remaining silent toward non-plastic particles. Emerging MIP-based optical sensors demonstrate that high polymer specificity can be achieved even in chemically simple substrates [59].

### 3.1. 4-Dimethylamino-4′-nitrostilbene (DANS): Achieving “Rainbow” Differentiation Based on Polarity Differences

4-Dimethylamino-4′-nitrostilbene (DANS) exemplifies a sophisticated solvatochromic probe that directly reports polymer polarity to identify microplastics (MPs). Its molecular structure, featuring a strong donor-acceptor system (Figure 4A), gives rise to a dipolar excited state that is stabilized in polar matrices, resulting in a pronounced emission redshift. This fundamental mechanism allows DANS to translate subtle polarity differences among plastics into distinct fluorescence colors. In practice, a straightforward staining protocol effectively labels prevalent MPs, including PP, LDPE, HDPE, PS, and PET. Under UV illumination, the stained particles emit an intuitive “MP rainbow” (Figure 4C)—from blue (non-polar PP) to red (polar PET)—enabling rapid, naked-eye screening.

The polarity-dependent behavior is quantitatively validated by steady-state fluorescence spectroscopy, where polyolefins (PP, PE) show peaks at 450–520 nm, while more polar polymers (PS, PET) exhibit peaks above 520 nm (Figure 4B). Notably, DANS’s sensitivity resolves subtle structural differences, such as LDPE versus HDPE branching. To transcend qualitative observation and achieve quantitative discrimination in complex mixtures, spectral phasor analysis transforms pixel-level emission spectra into a 2D phasor plot (Figure 4D). This approach reveals distinct polymer clusters and enables automated, color-coded mapping for direct quantification of relative abundances in heterogeneous samples [41].

Despite its strengths, DANS’s discriminatory power can be constrained in highly complex mixtures containing aged polymers due to spectral overlap, and its requirement for elevated staining temperature may limit field applicability. Nonetheless, DANS robustly validates the feasibility of solvatochromism for chemical-aware MP recognition, paving the way for next-generation probes capable of polymer-specific “switch-on” signals to achieve unparalleled specificity.

However, the reliability of DANS’s polarity-dependent discrimination can be profoundly influenced by environmental aging processes, such as oxidative weathering, which alter the surface chemistry of microplastics. Prolonged exposure to UV radiation, atmospheric oxygen, and hydrolytic conditions introduces polar functional groups (e.g., hydroxyl, carbonyl, and carboxyl moieties) onto polymer surfaces, thereby increasing overall hydrophilicity and potentially homogenizing the solvatochromic response across diverse polymer types [60,61]. For instance, photooxidative degradation of polyolefins like PP and PE can shift their emission profiles toward those of inherently polar polymers such as PET, leading to spectral overlap and diminished “rainbow” resolution in aged samples [62]. This compromised selectivity underscores the need for complementary techniques, such as Fourier-transform infrared spectroscopy (FTIR), to verify surface modifications in real-world environmental matrices. Future probe iterations should incorporate adaptive designs, perhaps through dual-mode sensing, to mitigate these aging-induced artifacts and enhance long-term dependability in ecological monitoring.

The polarity-sensitive behavior of DANS originates from its strong intramolecular charge-transfer (ICT) character. The electron-donating dimethylamino group and the electron-withdrawing sulfonyl moiety create a dipolar excited state whose emission energy shifts markedly with microenvironment polarity, producing the characteristic “rainbow” fluorescence. Thus, variations in polymer surface polarity directly modulate the ICT stabilization and generate polymer-dependent emission colors.

Notably, fluorescence intensity and lifetime are not solely determined by probe–polymer interactions but also by microplastic morphology, including surface roughness, oxidation level, and porosity, all of which modify local polarity and non-radiative decay channels. Environmental matrix artifacts—such as dissolved organic matter (DOM), salinity, and microplastic-derived leachates—can further perturb spectral and lifetime signatures and reduce classification accuracy. These combined effects often undermine the stability of straightforward turn-on probes when applied to heterogeneous environmental samples. Consequently, ratiometric probes or lifetime-based readouts generally provide more robust quantification under complex matrices [63].

### 3.2. DPNA ((E)-N-(2-((4-(Diphenylamino)benzylidene)amino)phenyl)-7-nitrobenzo[c][1,2,5]oxadiazol-4-amine): A Breakthrough Probe Specifically Recognizing Polyurethane (PU)

The development of the DPNA probe marks a strategic shift from broad-spectrum sensing toward exclusive, turn-on recognition of specific polymers. DPNA achieves exceptional selectivity for polyurethane (PU) not through hydrophobicity, but via a sophisticated mechanism of hydrogen-bond-driven recognition. Its molecular structure is engineered to form multiple directional hydrogen bonds with urethane groups on the PU chain, a binding event that rigidifies the probe and restricts intramolecular rotation (RIR). This RIR process suppresses non-radiative decay, activating a strong near-infrared (NIR) fluorescence turn-on exclusively for PU [64].

The photophysical basis of this mechanism is rooted in DPNA’s pronounced solvatochromism (Figure 1). As shown in Figure 5A, its absorption is highly sensitive to environmental polarity, a correlation quantified by the Lippert–Mataga plot (Figure 5D). This property is key to its function: while DPNA remains non-fluorescent in polar aqueous environments (Figure 5C), it emits a potent NIR signal upon binding to PU (Figure 5B). The molecular design and electronic transitions underpinning this process are further detailed in Figure 6.

Practically, DPNA offers distinct advantages, including a dual-mode colorimetric and NIR fluorescence response. The visible color change enables rapid screening, while the NIR emission (~667 nm) minimizes background interference for sensitive quantification. Staining is exceptionally rapid, reaching maximum intensity within one minute. Its utility is demonstrated in complex environmental samples such as river water and sand, effectively identifying diverse PU forms (foams, films, fibers). The probe’s high selectivity is quantitatively illustrated in Figure 7A, which compares fluorescence intensity across different microplastics, and is visually confirmed by the corresponding micrographs (Figure 7B). Furthermore, DPNA enables crucial ecotoxicological insights, having been used to visually track the ingestion and accumulation of PU microplastics in the digestive tract of Daphnia magna.

In summary, DPNA establishes a new benchmark in rational probe design by moving beyond physical adsorption to true chemical recognition. Its combination of rapid, dual-mode detection, high specificity, and proven performance in complex environments provides a critical methodological foundation for monitoring PU contamination and investigating its long-term environmental and toxicological impacts [64]. Notwithstanding these advancements, the specificity of DPNA, which hinges on precise hydrogen-bonding interactions with urethane functional groups in PU, may be undermined by environmental oxidative processes that functionalize polymer surfaces. Aging phenomena, including photooxidation and thermal degradation, can introduce competing polar sites (e.g., peroxides, ketones, and aldehydes) on PU chains, potentially disrupting the probe’s restriction of intramolecular rotation (RIR) mechanism and leading to off-target fluorescence activation or attenuated signal intensity [62,65]. Empirical studies on weathered PU microplastics have demonstrated that such surface alterations can reduce binding affinity by up to 30–50%, resulting in false negatives or cross-reactivity with oxidized non-PU polymers like aged PET [66,67]. This vulnerability highlights a critical gap in probe robustness for field applications, where microplastics are invariably subjected to heterogeneous aging. To bolster dependability, integrating surface pretreatment protocols or developing aging-resistant analogs—perhaps via sterically shielded recognition motifs—will be essential for sustaining DPNA’s utility in longitudinal toxicological assessments and global contamination mapping.

Beyond DPNA, next-generation NIR-I (700–900 nm) and NIR-II (1000–1700 nm) fluorophores provide further reductions in biological autofluorescence and allow deeper optical penetration for real-time monitoring of NP internalization pathways. Donor–acceptor–donor cyanine derivatives, benzobisthiadiazole-based chromophores, and activatable NIR-II probes have demonstrated exceptional signal-to-background ratios in live-cell and in vivo imaging. These scaffolds also offer superior photostability and reduced off-target activation, features particularly valuable for tracking dynamic NP transport across membranes and into organelles. Recent NIR-II studies illustrate that subcellular trafficking can be visualized with significantly enhanced temporal and spatial resolution compared with conventional visible/NIR-I dyes [68].

For DPNA, the key recognition originates from the carbamate (–NH–COO–) unit, which serves as a hydrogen bond donor capable of selectively interacting with urethane linkages in PU. These directional hydrogen-bonding interactions constrain intramolecular rotation and trigger the turn-on response, accounting for the probe’s high selectivity toward polyurethane.

### 3.3. 1,3,6,8-Pyrene Tetrasulfonate (PTSA): Probes with Good Water Solubility But Potential Toxicity

1,3,6,8-Pyrene tetrasulfonate (PTSA) represents a distinct class of hydrophilic probes that offer practical advantages for detecting synthetic microfibers in aqueous environments. Unlike hydrophobic dyes requiring organic solvents, PTSA exhibits excellent water solubility and stability, enabling its use in field-deployable systems. It has been effectively paired with a portable photometer for the real-time quantification of microfibers from sources like surgical masks.

PTSA demonstrates a selective binding affinity for polymers such as polyethylene and polypropylene, which constitute the majority of disposable face masks. This interaction, confirmed by characteristic shifts in FT-IR spectra (Figure 8), forms the basis of its detection principle via fluorescence quenching. Optimal performance is achieved at neutral pH (~7.2), with a contact time of 2 h, and across a wide temperature range (20–60 °C), as detailed in the optimization study [69] (Figure 9).

The probe’s environmental relevance was confirmed in diverse real-world water samples, including river, estuarine, and seawater, where it maintained strong fluorescence signals. A critical assessment, however, revealed significant ecotoxicity: PTSA-bound microfibers caused 100% lethality in *Artemia salina* within 48 h. To mitigate this, an eco-friendly remediation strategy was developed, employing xanthan gum for competitive dye displacement followed by solar photodegradation, successfully reducing fluorescence intensity by over 97% (Figure 10). However, this underscores a broader challenge in probe design: balancing sensitivity with low biotoxicity. Emerging evidence links MPs/NPs themselves to carcinogenic risks via chronic inflammation and genotoxicity [26], making non-toxic probes essential for in vivo studies to avoid confounding toxicity data from the probe versus the MNP.

To fully leverage ML-driven classification based on phasor-transformed FLIM data, standardized data infrastructures and calibration protocols are urgently required. This includes community-shared FLIM benchmark datasets with verified polymer labels, harmonized lifetime calibration standards, and transparent reporting of model accuracy, calibration errors, and decision thresholds. Recent open-source phasor-processing frameworks demonstrate that cross-platform reproducibility is feasible when acquisition and preprocessing steps are rigorously documented. Establishing such shared pipelines would allow ML-based FLIM identification of MPs/NPs to transition from exploratory demonstrations to validated analytical workflows [70,71].

In summary, PTSA exemplifies a sensitive, water-compatible probe for decentralized microfiber monitoring, yet its pronounced ecotoxicity—manifested as complete lethality in model organisms like *Artemia salina*—highlights a fundamental tradeoff in probe development [69]. This underscores the urgent need for rigorous biocompatibility assessments, including chronic exposure assays in vertebrate models, to preclude confounding effects in toxicological endpoints such as oxidative stress or genotoxicity [72]. Moreover, incorporating biodegradability into probe architectures, such as through labile ester linkages or enzyme-responsive motifs, is essential to mitigate environmental persistence and enable sustainable deployment in long-term monitoring programs [73]. By prioritizing these criteria, future iterations can evolve from mere analytical tools to benign platforms for elucidating MNP-induced pathologies, including potential carcinogenic pathways [26].

### 3.4. Aggregation-Induced Emission (AIE) Probes: Harnessing Restricted Intramolecular Motion for Highly Sensitive Detection

Aggregation-induced emission (AIE) luminogens (AIEgens) have emerged as powerful tools for detecting microplastics (MPs) and nanoplastics (NPs), leveraging their unique photophysical property where fluorescence is weak in dispersed states but dramatically enhanced upon aggregation or restriction of intramolecular motion [74]. This “turn-on” mechanism contrasts with traditional fluorophores suffering from aggregation-caused quenching (ACQ), enabling high signal-to-noise ratios and photostability for sensitive MP/NP tracking in complex matrices [75]. In MP detection, AIEgens bind to polymer surfaces via hydrophobic and electrostatic interactions, triggering fluorescence enhancement proportional to plastic concentration [76,77].

For nanoplastics (NPs), AIEgens offer paradigm-shifting advantages over their use in microplastics (MPs) detection, directly countering NP-specific challenges such as aqueous aggregation and biological barrier penetration. Unlike MPs, which can be visualized via bulk staining, NPs’ tendency to form clusters in water often yields false positives in traditional fluorescence assays due to signal overlap; AIEgens mitigate this through their inherent ‘turn-on’ upon aggregation or motion restriction, transforming potential artifacts into quantifiable signals with picomolar sensitivity [78]. In biological contexts, where NPs permeate barriers like the blood–brain interface, AIE-labeled NPs enable real-time in vivo imaging in models such as zebrafish, revealing differential accumulation in neural tissues compared to MPs’ primarily gastrointestinal localization [79]. This exquisite selectivity stems from tunable hydrophobic/electrostatic interactions that restrict rotation more efficiently in NPs’ high-surface-area matrices, outperforming MP applications where size alone aids discrimination.

A key application involves NIR-II AIE fluorophore-labeled MPs/NPs for in vivo visualization in zebrafish, where labeled polystyrene particles (2 μm MPs and 100 nm NPs) accumulate heterogeneously in the gut, with MPs favoring fore- and mid-regions, achieving picomolar sensitivity without autofluorescence interference [79]. This method quantifies uptake and depuration kinetics, refining toxicokinetic models for environmental risk assessment [79] (Figure 11). Similarly, fluorescent molecular rotors like DCVJ exhibit microenvironment-dependent quantum yield shifts upon binding polystyrene NPs, enabling quantification down to 475 μg/L in water [80].

Peptide-AIEgen conjugates further enhance selectivity, where self-assembly restricts rotation for ALP detection but extends to polymer sensing via structural tuning [81] (Figure 12). In real samples, AIEgen@GO nanocomposites provide dual-stage turn-on for nucleic acids, adaptable to MP detection with pM sensitivity [82]. Advances in biomedical AIEgens highlight diagnostic potential, with high quantum yields for biomarker-like MP imaging [83] (Figure 13).

Advantages include rapid, wash-free assays with LODs as low as 5.08 pg/mL, outperforming traditional methods in reproducibility (2.6–9.8% CV) [84,85]. Challenges in NP quantification, such as size-induced aggregation leading to false positives and enhanced barrier penetration complicating in vivo localization, persist and diverge from MP hurdles; these necessitate machine learning integration for pattern recognition, alongside AIE’s motion-restriction mechanism to enhance specificity in dynamic biological matrices [86,87,88,89].

**Figure 12 ijms-26-11283-f012:**
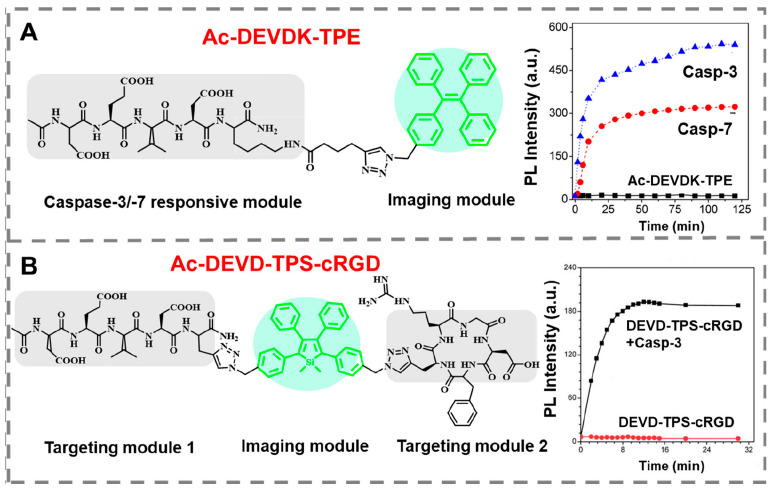
(**A**) Structure of the Ac-DEVDK-TPE probe (left) and its time-dependent PL intensity upon addition of caspase-3 and caspase-7 from 0 to 120 min (right). Blue triangles and red dots represent the PL intensity of Ac-DEVDK-TPE in the presence of caspase-3 and caspase-7, respectively, while black squares correspond to the probe alone without enzyme ([caspase-3] = [caspase-7] = 100 pM). (**B**) Structure of the Ac-DEVD-TPS-cRGD probe (left) and its PL intensity with and without caspase-3 over 30 min (right), where black squares indicate the probe incubated with caspase-3 and red dots indicate the probe in the absence of caspase-3 [90]. Copyright © 2019 American Chemical Society.

**Figure 13 ijms-26-11283-f013:**
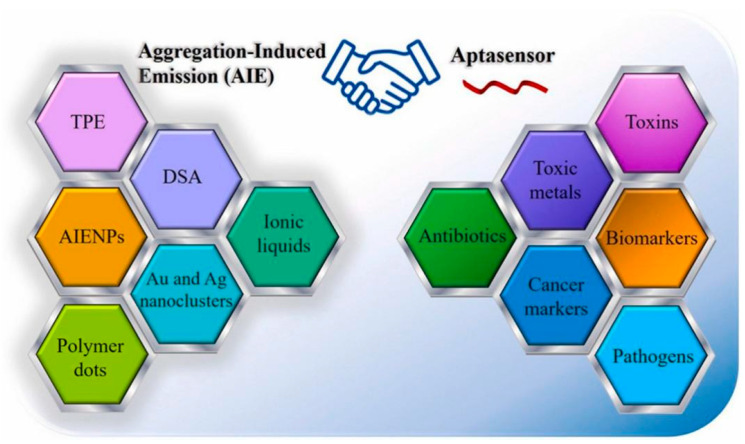
Schematic of AIE mechanism in aptasensors, showing aggregation-enhanced emission [74]. Copyright 2025 Elsevier B.V.

Future efforts should focus on portable, plasmonic-enhanced AIE platforms for field detection [90]. AIEgens are particularly promising for in vivo carcinogenicity studies, as they enable monitoring of MNP internalization and translocation in models like zebrafish, revealing bioaccumulation in metabolically active organs (e.g., liver, kidneys) and associated oxidative stress [26].

### 3.5. Fluorescence Resonance Energy Transfer (FRET)-Based Ratiometric Probes: Self-Calibrating Detection Through Energy Transfer

To address the inherent quantitative limitations of simple “turn-on” (single-intensity) probes, which are highly susceptible to fluctuations in instrumental factors (e.g., excitation lamp power, detector sensitivity), probe concentration, photobleaching, and matrix-induced quenching, ratiometric fluorescent probes have been engineered. These advanced systems provide built-in self-calibration for robust quantification. By measuring the ratio of fluorescence emission at two distinct wavelengths (e.g., *I_A_*/*I_B_*), which correlates with the analyte concentration, the probe effectively normalizes the signal against most analyte-independent variables. This ratiometric output thus affords superior accuracy and reliability for quantitative analysis, particularly in complex and heterogeneous environmental matrices, directly overcoming a primary drawback of straightforward turn-on systems. A predominant design strategy for ratiometric sensing employs Förster Resonance Energy Transfer (FRET), where a dual-dye molecular construct enables reversible FRET modulation. This is often seen in probes responsive to environmental cues like pH or viscosity, which can be altered by MP degradation products. For example, the near-infrared pH probe SHD-SiRho exhibits reversible ratiometric changes in its absorption (680/730 nm) and fluorescence (680/775 nm) signals upon pH cycling between 4.0 and 8.0 (Figure 14), providing a robust self-calibrated readout. The FRET process can also transition between on and off states during the absence and presence cycles of HS^−^ [91,92]. Similarly, gold nanocluster–rhodamine B assemblies detect hydrogen sulfide via surface-triggered FRET quenching, inspiring MP probes where polymer interactions regulate donor–acceptor distance, as “H_2_S could finely regulate surface structure of gold nanoclusters (AuNCs) through reduction in surface Au(I)-ligand motifs and further quench their fluorescence by a two-stage kinetic reaction process” [93] (Figure 15).

Ratiometric advantages include high sensitivity (limits down to 4.8 pM) and selectivity over interferents, with built-in calibration via dual-channel ratios (e.g., green/orange emissions), providing “a high energy transfer efficiency (99.6%)” [94]. These probes excel in visualizing MP distributions through pseudocolor ratiometric imaging, enabling differentiation of MP types by emission wavelength shifts (e.g., 476–595 nm) or intensity ratios, as demonstrated in cellular and zebrafish models where “the probe demonstrates the capability to reveal the dynamic distribution of HSO_3_^−^ levels across different regions in both living cells and zebrafish” [95]. In MP contexts, laser-induced fluorescence combined with PCA-SVM algorithms achieves 100% classification accuracy for marine MPs, suggesting FRET integration could enhance specificity, as “the classification accuracy can be increased to 100% by PCA combined with SVM and KNN algorithm” [96].

Applications span food safety monitoring (e.g., bisulfite in MP-contaminated samples) and in vivo tracking of MP uptake, with reversible redox responses aiding dynamic studies, as “this article provides a review of different FRET approaches for the ratiometric monitoring of the self-assembly and formation of nanoparticles, their in vivo fate, integrity and drug release” [97]. Overall, FRET ratiometric probes provide robust, artifact-free MP detection, with future designs focusing on MP-specific recognition motifs for enhanced environmental and biomedical utility, as “our probe was successfully utilized in ratiometric photoacoustic and fluorescence tumor imaging and ratiometric fluorescence imaging-guided tumor resection” [98].

### 3.6. Considerations for Physical Parameters and Standard Sample Selection in MNP Detection

Beyond the chemical and structural aspects of fluorescent probes, physical parameters such as particle size (diameter), shape, surface roughness, and density play a critical role in influencing detection performance. For instance, nanoplastics (<100 nm) are prone to aggregation in aqueous environments due to van der Waals forces, which can lead to signal overlap, false positives, or reduced sensitivity in fluorescence-based assays [28,29]. In contrast, larger microplastics (>1 μm) may exhibit uneven probe adsorption owing to lower surface area-to-volume ratios, affecting quantification accuracy. Environmental aging further exacerbates these issues by introducing surface modifications (e.g., oxidation leading to increased roughness or polarity), which can alter probe binding mechanisms like solvatochromism or restriction of intramolecular rotation (RIR) [68,69,70].

To address these challenges, selecting appropriate standard samples is essential for method validation, calibration, and reproducibility. Standard samples should mimic real-world MNPs in terms of physical properties, with certified particle size distribution (e.g., via dynamic light scattering, DLS), monodispersity (polydispersity index, PDI < 0.2), and chemical purity (>95%). Commercial sources include polystyrene (PS) microspheres from Sigma-Aldrich (St. Louis, MO, USA) or reference materials from NIST (NIST, Gaithersburg, MD, USA; e.g., SRM 2881). The choice of standards must align with the detection technique’s resolution and the target matrix (e.g., water, soil, or biological tissues). Table 1 summarizes guidelines for selecting standards tailored to key techniques discussed in this review.

General principles for standard selection include: (i) matching particle size to technique resolution (e.g., >1 μm for fluorescence microscopy; 50 nm−10 μm for flow cytometry); (ii) simulating environmental conditions with aged samples (UV/oxidation treatment); (iii) using ISO/TS guidelines for sampling consistency; and (iv) integrating machine learning (e.g., PCA−SVM) to compensate for physical variability, improving accuracy to >95%. By incorporating these considerations, future probe designs can achieve greater robustness in complex matrices, bridging the gap between laboratory validation and field applications.

## 4. Cutting-Edge Detection System: Signal Enhancement Technology Beyond the Probe Itself

To overcome sensitivity limitations in detecting trace-level microplastics, the field is increasingly leveraging signal amplification strategies from materials science and nanotechnology. This chapter explores three complementary approaches that enhance detection without solely relying on probe chemistry: one that eliminates the need for staining, one that amplifies existing stains, and one that uses “smart” probes with built-in enhancement.

### Plasmon-Enhanced Fluorescence (PEF): Visualizing “Dye-Free” Microplastics

Plasmon-Enhanced Fluorescence (PEF) enables the direct visualization of unlabeled microplastics by dramatically amplifying their intrinsic autofluorescence. This technique utilizes nanostructured substrates, such as gold nanopillars, which generate a localized electromagnetic field via surface plasmon resonance. A pioneering application of this principle achieved a transformative ~70-fold signal enhancement, pushing detection limits to the femtogram level (0.35 fg) and allowing visualization of particles beyond the resolution of conventional microscopy [99].

The efficacy of this platform was demonstrated by detecting diverse microplastics, including LDPE and biodegradable PBAT fibers. The superhydrophobic nature of the gold nanopillar substrate concurrently concentrates particles from aqueous samples, ensuring their immobilization within the enhanced field for optimal signal amplification, as quantitatively detailed in Figure 16 [99,100]. This “dye-free” approach is vital for samples where staining is undesirable, though its reliance on sophisticated fabrication may currently limit widespread deployment.

While PEF and MEF systems achieve unparalleled sensitivity through plasmonic amplification, their operational complexity—requiring nanofabricated substrates and specialized instrumentation—incurs high expenses (e.g., >$10,000 per setup) and limits scalability for on-site monitoring [101]. In contrast, low-cost alternatives like portable fluorescence spectrometers or smartphone-integrated readers for basic probes (e.g., Nile Red) enable rapid field deployment at <$500, with high performance in throughput but lower resolution for trace NPs. Balancing these, hybrid approaches integrating simplified PEF metasurfaces with cost-effective optics could reduce expenses by 50–70% while maintaining femtogram-level detection, facilitating broader environmental surveillance.

In contrast to microplastics (MPs), where PEF primarily amplifies autofluorescence for surface-level detection, its application to nanoplastics (NPs) addresses more intricate challenges, including aggregation-driven false positives in water and penetration of biological barriers. NPs’ sub−100 nm scale often results in signal dilution from clustering, but PEF’s localized plasmonic fields concentrate electromagnetic enhancement at the nanoscale, enabling femtogram-level resolution and deconvolution of aggregates without staining [99]. For biological penetration, PEF-integrated metasurfaces facilitate subcellular imaging of NPs crossing barriers like the intestinal epithelium, revealing toxicodynamic pathways such as oxidative stress induction, which are less pronounced in MPs due to their limited invasiveness. This differential enhancement underscores PEF/MEF’s versatility, with MEF further boosting stained NPs via metal-fluorophore coupling for in vivo carcinogenicity studies.

## 5. Conclusions

The field of fluorescent probing for micro- and nanoplastic (MNP) identification has undergone a remarkable evolution, transitioning from rudimentary staining techniques towards sophisticated, mechanism-driven detection systems. This journey began with foundational strategies relying on non-specific hydrophobic adsorption, exemplified by widely used dyes like Nile Red, Coumarin 6, and Rhodamine B (Table 2). While these first-generation probes provided a crucial, cost-effective, and accessible entry point for rapid MNP screening, their inherent limitations—primarily lack of specificity, susceptibility to false positives from natural organics, dye leaching, and in some cases, significant biotoxicity—highlighted an urgent need for more advanced methodologies.

This impetus drove the development of a new generation of probes designed for precise identification. The solvatochromic dye DANS represented a significant leap forward, enabling a “microplastic rainbow” differentiation based on polymer polarity. However, the true paradigm shift arrived with probes like DPNA, which abandoned the concept of general affinity in favor of exquisite specificity. By leveraging multiple hydrogen bonds to selectively target polyurethane (PU) and employing a restriction of the intramolecular rotation (RIR) mechanism for a “turn-on” near-infrared response, DPNA established a new benchmark for polymer-specific detection. Concurrently, the case of PTSA underscored the critical, non-negotiable importance of environmental safety in probe design, reminding the community that high performance must be coupled with low ecotoxicity.

Pushing the boundaries of sensitivity even further, the field has begun to integrate advanced signal enhancement technologies that operate beyond the probe’s intrinsic properties. Techniques like Plasmon-Enhanced Fluorescence (PEF) and Metal-Enhanced Fluorescence (MEF) harness the power of nanostructured metals to amplify signals by orders of magnitude. PEF allows for the direct visualization of unlabeled microplastics by enhancing their weak autofluorescence, while MEF dramatically boosts the signal from stained particles, enabling detection at femtogram levels. Simultaneously, the advent of “smart” probe systems, such as fluorogenic hyaluronic acid nanogels, offers an elegant solution to false positives from dye aggregation. Their “activate-on-target” mechanism and compatibility with fluorescence lifetime imaging (FLIM) provide a powerful combination of high signal-to-noise ratio and the potential for polymer discrimination.

The clear trajectory for the future lies in the strategic convergence of these parallel advancements. The ultimate goal is the creation of integrated detection platforms that marry the unparalleled specificity of advanced chemical probes like DPNA with the extreme sensitivity afforded by physical enhancement substrates like plasmonic metasurfaces. To enhance scientific impact, future probe development must prioritize applications in toxicological research, particularly carcinogenesis. Evidence suggests that MPs/NPs contribute to human cancer risk through toxicokinetic pathways (absorption via ingestion, inhalation, and dermal routes) and toxicodynamic mechanisms (e.g., DNA damage, cellular stress, inflammation, and transcription factor modulation). Probes like AIE luminogens and PEF/MEF platforms are particularly adept at resolving NP-specific challenges, such as aggregation-induced artifacts and barrier penetration, by providing aggregation-resistant ‘turn-on’ signals and plasmonic amplification, respectively, thereby enabling differential analysis from MPs and advancing mechanistic insights into oncogenesis. Advanced probes like AIEgens, NIR emitters, and DPNA enable real-time, in vitro and in vivo monitoring of MNP internalization (via endocytosis, phagocytosis), bioaccumulation in organs (e.g., brain, lungs, gut), and resultant cellular damage, transforming probes from mere detection tools into instruments for resolving whether MNPs are oncogenic agents. This addresses key gaps, such as longitudinal epidemiology and human-relevant models (organoids, organs-on-chips), while minimizing probe biotoxicity to avoid confounding results.

A pivotal yet underexplored facet of probe evolution is the mitigation of inherent toxicity, which must be elevated from a peripheral concern to a foundational design principle. Current probes, while efficacious, often exhibit cytotoxicity that compromises their utility in protracted in vivo exposure paradigms, potentially obfuscating MNP-specific toxicodynamic responses such as chronic inflammation or DNA adduct formation [26,102]. To surmount this, next-generation probes should adhere to stringent biocompatibility criteria, including hemocompatibility assays, minimal immunogenic potential, and negligible interference with cellular redox homeostasis, as validated in organoid or organ-on-chip models [103]. Concurrently, biodegradability must be engineered via cleavable moieties (e.g., peptide or glycosidic bonds) that facilitate metabolic clearance, thereby averting bioaccumulation and enabling iterative dosing in longitudinal studies [104]. These attributes are indispensable for scaling to extensive environmental monitoring, where probes must persist in ecosystems without exacerbating contamination or eliciting non-target effects in sentinel species [105]. By embedding green chemistry tenets—such as atom economy and renewable feedstocks—into molecular blueprints, the field can yield probes that are not only sensitive and specific but also ecologically innocuous, thus bridging the chasm between analytical prowess and sustainable application [106].

Future research must focus on several key challenges: expanding the ‘toolkit’ of specific probes to cover a wider range of common polymers beyond PU while accounting for environmental aging effects on surface chemistry that may compromise probe specificity; drastically reducing the cost and complexity of enhancement substrates to enable widespread deployment; rigorously validating these next-generation systems in complex, real-world environmental and biological matrices, including aged polymer samples; integrating biocompatibility and biodegradability as core metrics through standardized toxicity profiling; and prioritizing the principles of green chemistry to ensure all new probes and materials are inherently non-toxic, biodegradable, and environmentally benign.

In illuminating the invisible world of MNPs, fluorescent probes have proven to be indispensable emerging tools. Their continued refinement and integration into smart, sensitive, and specific detection systems are paramount. These advanced tools will provide the critical data needed to accurately map MNP contamination, elucidate their environmental transport and fate, assess their ecological impacts with greater precision, and ultimately, evaluate the potential risks they pose to human health, thereby informing effective mitigation and policy strategies for this pervasive environmental challenge.

## Data Availability

No new data were created or analyzed in this study. Data sharing is not applicable to this article.

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
