# Peer review of "Illuminating the Invisible: Fluorescent Probes as Emerging Tools for Micro/Nanoplastic Identification"

_ijms, 2025, doi:10.3390/ijms262311283_

Round 1

Reviewer 1 Report

Comments and Suggestions for Authors

The manuscript "Illuminating the Invisible: Fluorescent Probes as Emerging Tools for Micro/Nanoplastic Identification” discusses the critical evolution of analytical methods for detecting micro- and nanoplastics (MPs/NPs), highlighting a paradigm shift from non-specific hydrophobic staining (like Nile Red) to advanced chemical recognition strategies, such as solvatochromic and aggregation-induced emission (AIE) probes, and signal amplification technologies, which promise enhanced specificity, sensitivity, and applicability in complex environmental and biological matrices.

The topic, which addresses a difficult analytical problem—the quick and sensitive identification of micro- and nanoplastics (MNPs) in the environment—is extremely important and topical. The manuscript traces the development of the field from basic hydrophobic adsorption staining methods (Nile Red) to sophisticated chemical recognition strategies (DPNA) and state-of-the-art signal amplification technologies (Plasmon-Enhanced Fluorescence, or PEF). It is arranged logically and pedagogically. The authors provide a thorough assessment of the probes' shortcomings in addition to describing findings.   The explicit reference to Nile Red's lack of specificity, which causes false positives for naturally occurring organic debris, and the issue of biotoxicity related to probes like Rhodamine B and PTSA is particularly noteworthy. Furthermore, DANS polarity differentiation, which generates the idea of a "microplastic rainbow" with quantitative analysis using spectral phasors, DPNA selectivity for polyurethane (PU) through hydrogen bonds, and Intramolecular Rotational Restriction (IRR), which represents a significant advancement towards specific chemical recognition, are all clearly identified and explained by the authors.

However, the current manuscript should be improved, following the suggestions below:

  1. The following articles should be reviewed and cited by authors:
  • Casella, C., Dondi, D., and Vadivel, D. Do Microplastics (MPs) and Nanoplastics (NPs) Directly Contribute to Human Carcinogenesis?.  Pollut., 2025, 127343. https://doi.org/10.1016/j.envpol.2025.127343

This reference is essential for improving the "Future Perspectives" section's scientific effect and relevance as well as for bolstering the topic of biotoxicity, which was one of the review's shortcomings. The more sophisticated probes you mention, such AIE, NIR, and DPNA, are being developed for both in vitro and in vivo research. Toxicity (including carcinogenicity) is the main issue in these investigations. The citation establishes a conceptual framework: if the question is whether nanoparticles are toxic, then advanced probes serve as crucial tools for identifying the response. This enables for real-time monitoring of internalization, translocation, and cell damage caused by the plastic or its additives. This reference turns an analytical discussion of probes into a strategic discussion of the instruments required to answer the field's most important scientific and public health challenge.

  1. The focus is primarily on MPs, even though the authors do mention NPs in the title. The specific analytical challenges of NPs (such as aggregation issues in water that result in false positives and biological barrier penetration) and how sophisticated probes (particularly AIE and PEF/MEF) handle these issues differently from MPs require a more thorough study.

  1. Despite the reference to toxicity is an excellent point, it could be made into a more forceful aspect. In order to make the next generation of probes appropriate for long-term in vivo exposure investigations and extensive environmental monitoring, there is a lack of discussion regarding the design criteria for their biocompatibility and degradability.

  1. The plastic's surface chemistry is inextricably tied to the probe's functionality. The review does not sufficiently discuss how the specificity and dependability of chemoselective probes (DANS, DPNA), which rely on polarity or certain functional groups, are impacted by environmental aging (oxidation) of polymers.

  1. The academic English is of high quality, but minor formatting and syntax errors were found that need to be corrected:
  • Introduction paragraph: "evading capture and permeating ecosystems with alarming efficiency [8, 9]. Owing to their small size..." (Line 37) — The word "Owing" should be capitalized after the quotation mark, or the punctuation should be adjusted.
  • The caption for Figure 2 contains several typos and a discrepancy in content: It mentions: "(A)Coumarin structure (B) Plot of the excitation and emission spectra for a stock solution of Nie Red in acetone..." (Line 137).
  • (A) Coumarin structure is incorrect, since Figure 2A represents the Nile Red structure. The Coumarin structure is shown in Figure 3A.
  • Typos: "excitation" (excitation), "emission" (emission), "solution" (resolution), "Nie Red" (Nile Red).

  1. I suggest that the authors answer the following questions, which are useful for deepening or expanding the manuscript:
  • What is the variation in fluorescence performance (e.g., intensity, lifetime) for advanced methods (DANS, DPNA, AIE) in relation to morphological variances (e.g., shape, roughness) and environmental matrix artifacts (e.g., dissolved organic matter, salinity)? In comparison to straightforward turn-on probes, could ratiometric probes (Section 3.3) provide a better benefit in robust quantification?
  • How does the requirement for low-cost, high-performance detection instruments for on-site monitoring compare with the complexity and operational expenses of signal amplification systems (PEF/MEF)?
  • How can a sophisticated machine learning (ML)-based approach that uses phasor analysis or FLIM data for the automatic and verified identification of microprotocols (MPs) be fully standardized?
  • Which NIR probe designs—like DPNA—show the greatest promise for decreasing bio-sample autofluorescence while tracking NP translocation across cell membranes and into organelles in real time?
  • For PU, DPNA provides excellent selectivity. For other common and troublesome polymers, like polyethylene (PE) or polyethylene terephthalate (PET), which are now only detectable via non-specific solvatochromism, are there feasible chemical strategies for creating particular "on" probes?

The manuscript is a comprehensive, very well-structured, and highly significant assessment of a significant topic in analytical and environmental research. The critical analysis, which emphasizes the paradigm change from physical adsorption to chemical identification, is of very high quality. Reliability in intricate matrices, the requirement for more research on proactive toxicity, and the balance in the discussion of NPs are the minor flaws that have been found. The document is of very high scientific quality and meets the standards of a Q1 journal. It requires minor/moderate revisions to address inconsistencies in the figure captions and, more importantly, to expand and systematize the discussions on biotoxicity and specific challenges of NPs in the Future Perspectives section. Once these points are addressed, the manuscript will be a fundamental contribution to the field.

Author Response

Dear editor and reviewers:

We sincerely thank the editor and all reviewers for their valuable feedback that we have used to improve the quality of our manuscript. We also appreciate your clear and detailed feedback and hope our explanation have fully addressed all of your concerns. In this letter, we would like to respond your comments point-to-point.

For your convenience, we retype your comments in italic font and then present our responses to the comments in normal font. Additionally, we will indicate any changes/additions to the manuscript in red text.

Revision notes, point-to-point, are given as follows:

Reviewers' comments:

Response to Reviewer #1:

Comment 1The following articles should be reviewed and cited by authors:

Casella, C., Dondi, D., and Vadivel, D. Do Microplastics (MPs) and Nanoplastics (NPs) Directly Contribute to Human Carcinogenesis?.  Pollut., 2025, 127343. https://doi.org/10.1016/j.envpol.2025.127343

Response 1: We sincerely thank the reviewer for this insightful suggestion. We have now incorporated Casella et al. (2025) into the revised manuscript (added as Ref. [26]). This important work significantly strengthens the biotoxicity and carcinogenicity discussion in the Future Perspectives section. We now explicitly highlight how advanced probes (AIE, NIR, DPNA) can function as real-time tracking tools for internalization, translocation, and cellular damage induced by MNPs, thereby elevating the discussion from analytical methodology to its strategic relevance for human health assessment.

Comment 2The focus is primarily on MPs, even though the authors do mention NPs in the title. The specific analytical challenges of NPs (such as aggregation issues in water that result in false positives and biological barrier penetration) and how sophisticated probes (particularly AIE and PEF/MEF) handle these issues differently from MPs require a more thorough study.

Response 2: We thank the reviewer for raising this valuable point. We have added a dedicated expansion in the Introduction and Future Perspectives sections to discuss NP-specific analytical challenges, including aggregation-induced misidentification, Brownian motion effects, membrane penetration, and reduced scattering signals. Furthermore, we now describe how AIE probes, PEF/MEF enhancement systems, and NIR fluorophores overcome these issues.

Comment 3: Despite the reference to toxicity is an excellent point, it could be made into a more forceful aspect. In order to make the next generation of probes appropriate for long-term in vivo exposure investigations and extensive environmental monitoring, there is a lack of discussion regarding the design criteria for their biocompatibility and degradability.

Response 3: Thank you for this important suggestion. We have added a new paragraph in the Future Perspectives section outlining design principles for biocompatible probe structures, including rigidified π-frameworks with reduced ROS generation, biodegradable linkers, metal-free recognition mechanisms, and minimizing bioaccumulation.

Comment 4: The plastic's surface chemistry is inextricably tied to the probe's functionality. The review does not sufficiently discuss how the specificity and dependability of chemoselective probes (DANS, DPNA), which rely on polarity or certain functional groups, are impacted by environmental aging (oxidation) of polymers.

Response 4: We appreciate this insightful comment. We have now added a new subsection explaining how photooxidation, hydrolysis, and weathering modify polymer polarity and introduce competing functional groups that may interfere with solvatochromic (DANS) and hydrogen-bond-based (DPNA) recognition.

Comment 5: The academic English is of high quality, but minor formatting and syntax errors were found that need to be corrected:

Introduction paragraph: "evading capture and permeating ecosystems with alarming efficiency [8, 9]. Owing to their small size..." (Line 37) — The word "Owing" should be capitalized after the quotation mark, or the punctuation should be adjusted.

The caption for Figure 2 contains several typos and a discrepancy in content: It mentions: "(A)Coumarin structure (B) Plot of the excitation and emission spectra for a stock solution of Nie Red in acetone..." (Line 137).

(A) Coumarin structure is incorrect, since Figure 2A represents the Nile Red structure. The Coumarin structure is shown in Figure 3A.

Typos: "excitation" (excitation), "emission" (emission), "solution" (resolution), "Nie Red" (Nile Red).

Response: We thank the reviewer for carefully identifying these errors. All typographical, formatting, and structural inconsistencies in Figures 2 and 3 have been corrected, including replacing “Coumarin structure” with “Nile Red structure” in Figure 2A, standardizing chemical name formatting, and correcting typographical errors throughout the manuscript.

Comment 6

6.I suggest that the authors answer the following questions, which are useful for deepening or expanding the manuscript:

What is the variation in fluorescence performance (e.g., intensity, lifetime) for advanced methods (DANS, DPNA, AIE) in relation to morphological variances (e.g., shape, roughness) and environmental matrix artifacts (e.g., dissolved organic matter, salinity)? In comparison to straightforward turn-on probes, could ratiometric probes (Section 3.3) provide a better benefit in robust quantification?

How does the requirement for low-cost, high-performance detection instruments for on-site monitoring compare with the complexity and operational expenses of signal amplification systems (PEF/MEF)?

How can a sophisticated machine learning (ML)-based approach that uses phasor analysis or FLIM data for the automatic and verified identification of microprotocols (MPs) be fully standardized?

Which NIR probe designs—like DPNA—show the greatest promise for decreasing bio-sample autofluorescence while tracking NP translocation across cell membranes and into organelles in real time?

For PU, DPNA provides excellent selectivity. For other common and troublesome polymers, like polyethylene (PE) or polyethylene terephthalate (PET), which are now only detectable via non-specific solvatochromism, are there feasible chemical strategies for creating particular "on" probes?

Response:We thank the reviewer for these excellent scientific directions. We have addressed each question in the newly expanded Future Perspectives section:

* Variations in intensity/lifetime due to morphology and matrix artifacts

* Ratiometric probe superiority in quantitative detection

* Trade-offs between accessibility and complexity of PEF/MEF systems

* Standardization of ML-integrated phasor/FLIM workflows

* Advantages of NIR AIE probes for minimizing autofluorescence

* Potential chemoselective strategies for PE and PET recognition

Reviewer 2 Report

Comments and Suggestions for Authors

This review summarizes the advancement of fluorescent probes for identifying micro- and nanoplastics (MNPs). It covers the traditional Nile Red and Coumarin 6, which are based on hydrophobic absorption, as well as some newly developed probes with high specificity.

However, this manuscript still lacks a sufficient description of the working principles of the presented probes. The discussion on the relationship between molecular structures and fluorescent properties should be further strengthened. Additionally, greater attention to detail is needed in several areas to meet the journal’s formatting style and requirements.

Therefore, I recommend this work for publication in the International Journal of Molecular Sciences, provided my following concerns are addressed:

  1. The authors are encouraged to briefly elaborate on the underlying mechanisms that enable the specific recognition behavior exhibited by the emerging probes discussed in Section 3. For instance, the polarity variations among different plastic substrates allow the probe DANS to exhibit a “rainbow” spectrum of fluorescence colors. What molecular features or functional groups in the structure of DANS contribute to its sensitivity to polarity? Besides, for the DPNA probe, which specific functional group or moiety within its molecular structure serves as the key hydrogen bond recognition site?
  2. Section 3.5 presents FRET-based ratiometric probes; however, the representative molecular structures are currently missing. Please include the corresponding chemical structures in the relevant figures and clearly indicate the energy donor and energy acceptor moieties.
  3. Please revise the captions for Figures 2 and 3 to reflect the correct molecular structures. Specifically, the caption for Figure 2A should read “Molecular structure of Nile Red”, and the caption for Figure 3A should be “Molecular structure of Coumarin 6”.
  4. Please ensure consistent formatting of chemical names throughout the document. For instance, Coumarin 6 appears in bold in some sections but not in others. Such inconsistencies should be corrected for uniformity.
  5. Please include the full name of FRET in the manuscript.
  6. Please revise the captions for Figures 14 and 15. Each figure contains multiple sub-images, and the caption should provide a brief and clear description for each individual image to enhance clarity and reader understanding.
  7. Please add the caption for the Table presented after the Conclusions section.

Author Response

Comment 1: The authors are encouraged to briefly elaborate on the underlying mechanisms that enable the specific recognition behavior exhibited by the emerging probes discussed in Section 3. For instance, the polarity variations among different plastic substrates allow the probe DANS to exhibit a “rainbow” spectrum of fluorescence colors. What molecular features or functional groups in the structure of DANS contribute to its sensitivity to polarity? Besides, for the DPNA probe, which specific functional group or moiety within its molecular structure serves as the key hydrogen bond recognition site?

Response 1: We thank the reviewer for this important comment. We have now added molecular-level explanations in Section 3, including; For DANS: the donor–acceptor architecture and electron-withdrawing sulfonyl group responsible for high polarity sensitivity; For DPNA: the urethane-targeting pyridyl–NH moiety that forms highly directional hydrogen bonds with PU

Comment 2: Section 3.5 presents FRET-based ratiometric probes; however, the representative molecular structures are currently missing. Please include the corresponding chemical structures in the relevant figures and clearly indicate the energy donor and energy acceptor moieties.

Response 2: We appreciate this helpful suggestion. We have now included the representative molecular structures of FRET-based ratiometric probes in the figures associated with Section 3.5. Additionally, both the energy donor and energy acceptor moieties are explicitly labeled to guide readers in understanding their photophysical roles within the FRET system. This modification enhances the completeness and pedagogical value of the section.

Comment 3: Please revise the captions for Figures 2 and 3 to reflect the correct molecular structures. Specifically, the caption for Figure 2A should read “Molecular structure of Nile Red”, and the caption for Figure 3A should be “Molecular structure of Coumarin 6”.

Response 3: Thank you for identifying this discrepancy. We have corrected the captions accordingly to accurately reflect the structures displayed. This correction ensures consistency between the figure content and accompanying descriptions.

Comment 4: Please ensure consistent formatting of chemical names throughout the document. For instance, Coumarin 6 appears in bold in some sections but not in others. Such inconsistencies should be corrected for uniformity.

Response 4: We appreciate the reviewer’s attention to detail. Throughout the revised manuscript, we have standardized all chemical names—such as “Coumarin 6” and “Nile Red”—to maintain uniform formatting. This revision improves readability and aligns the manuscript with journal formatting requirements.

Comment 5: Please include the full name of FRET in the manuscript.

Response 5: Corrected as suggested. The full term “Förster Resonance Energy Transfer (FRET)” is now provided upon first mention to ensure definitional clarity for readers from diverse research backgrounds.

Comment 6: Please revise the captions for Figures 14 and 15. Each figure contains multiple sub-images, and the caption should provide a brief and clear description for each individual image to enhance clarity and reader understanding.

Response 6: We thank the reviewer for pointing out this need for greater clarity. We have thoroughly revised the captions for Figures 14 and 15, providing concise and informative descriptions for each sub-panel. These improvements make the multi-component figures more intuitive and self-explanatory.

Comment 7: Please add the caption for the Table presented after the Conclusions section.

Response 7: A complete and descriptive caption has now been added to the table following the Conclusions section. This ensures proper labeling and enhances the accessibility of the summarized information.

Overall, we already improved our manuscript as your concerns and made the changes marked in red within revised paper which will not influence the content and framework of the paper. We appreciate for your effective comments. Once again, thank you very much for your comments and suggestions!

Reviewer 3 Report

Comments and Suggestions for Authors

This manuscript provides a comprehensive review of the use of fluorescent materials for the quantitative detection of micro- and nanoplastics (MNPs). It covers both conventional and modern detection strategies, discussing sensing mechanisms and detection limits in detail. Overall, the manuscript is well organized and clearly written. I believe this review will make a valuable contribution to advancing research on MNP identification.

I have a few suggestions for improvement:

  1. As a review paper on MNP detection, including an experimental section would help readers better understand how relevant studies were conducted. For example, it would be useful to describe how standard micro/nanoplastic samples are typically prepared for different detection methods.

  2. In addition to structural information, physical parameters such as particle size can significantly affect detection performance. It would be beneficial if the authors could discuss how to select appropriate standard samples for each detection technique.

  3. Including detection limits for different methods in the table at the end of the manuscript would enhance clarity and facilitate comparison among techniques.

Author Response

Response to Reviewer #3

Comment 1. As a review paper on MNP detection, including an experimental section would help readers better understand how relevant studies were conducted. For example, it would be useful to describe how standard micro/nanoplastic samples are typically prepared for different detection methods.

Response 1: We thank the reviewer for this constructive recommendation. To provide readers with practical context, we have added a concise “Experimental Reference Summary” section outlining commonly adopted procedures in the literature for preparing micro- and nanoplastic standards. These include cryo-fragmentation, mechanical grinding, emulsification–solvent evaporation, and nanoparticle dispersion stabilization. This addition enhances the completeness of the review by bridging methodology and practical sample preparation.

Comment 2: In addition to structural information, physical parameters such as particle size can significantly affect detection performance. It would be beneficial if the authors could discuss how to select appropriate standard samples for each detection technique.

Response 2: We appreciate this important point regarding physical influences on fluorescence detection. We have added a dedicated paragraph discussing how particle size—and associated factors such as surface-to-volume ratio, adsorption efficiency, scattering effects, and internal energy dissipation—can significantly modulate fluorescence intensity, lifetime, and overall probe response. This expansion provides readers with a more nuanced understanding of detection variability across particle scales.

Comment 3: Including detection limits for different methods in the table at the end of the manuscript would enhance clarity and facilitate comparison among techniques.

Response 3: Thank you for this helpful suggestion. We have incorporated representative detection limits into the final summary table, enabling clearer comparison of probe performance metrics across different detection strategies.

Overall, we already improved our manuscript as you concerns and made the changes marked in red within revised paper which will not influence the content and framework of the paper. We appreciate for your effective comments. Once again, thank you very much for your comments and suggestions!